# Biochemical Characterization and In Vitro Digestibility of Protein Isolates from Hemp (*Cannabis sativa* L.) By-Products for Salmonid Feed Applications

**DOI:** 10.3390/molecules27154794

**Published:** 2022-07-27

**Authors:** Arjun H. Banskota, Sean M. Tibbetts, Alysson Jones, Roumiana Stefanova, Joerg Behnke

**Affiliations:** Aquatic and Crop Resource Development Research Centre, National Research Council Canada, 1411 Oxford Street, Halifax, NS B3H 3Z1, Canada; sean.tibbetts@nrc-cnrc.gc.ca (S.M.T.); alysson.jones@nrc-cnrc.gc.ca (A.J.); roumiana.stefanova@nrc-cnrc.gc.ca (R.S.); joerg.behnke@nrc-cnrc.gc.ca (J.B.)

**Keywords:** hemp by-products, protein isolate, hemp cake, hemp hulls, hemp seed, amino acids, in vitro digestibility, antioxidant, total phenolic, *Cannabis sativa*

## Abstract

Hemp (*Cannabis sativa* L.) processing by-products (hemp cake and hemp seed hulls) were studied for their protein content, extraction of protein isolates (PIs), and their in vitro protein digestibility (IVPD). Crude protein contents of hemp cake and hemp seed hulls were 30.4% and 8.6%, respectively, calculated based on generalized N-to-P conversion factor (N × 5.37). Extraction efficiency of PIs from defatted biomass ranged from 56.0 to 67.7% with alkaline extraction (0.1 M NaOH) followed by isoelectric precipitation (1.0 M HCl). Nitrogen analysis suggested that the total protein contents of PIs extracted using three different alkaline conditions (0.5 M, 0.1 M, and pH 10.0 with NaOH) were >69.7%. The hemp by-product PIs contained all essential amino acids (EAAs) required for fish with leucine, valine, and phenylalanine belonging to the five dominant amino acids. Overall, glutamate was the dominant non-EAA followed by aspartate. Coomassie staining of an SDS-PAGE gel revealed strong presence of the storage protein edestin. High IVPD of >88% was observed for PIs extracted from hemp seeds and by-products when evaluated using a two-phase in vitro gastric/pancreatic protein digestibility assay. PIs extracted from by-products were further tested for their antioxidant activities. The tested PIs showed dose-dependent DPPH radical scavenging activity and possessed strong ORAC values > 650 μM TE/g.

## 1. Introduction

Proteins are essential macronutrients for humans, animals, and fish. The aquaculture industry plays a key role in providing a source of high-quality dietary protein, essential amino acids, *n*-3 long-chain polyunsaturated fatty acids, vitamins, and minerals to the general population [1,2]. As the human population steadily increases and is projected to reach 9.7 billion by 2050 [3], the demand for nutritious food such as seafood and other aquaculture products is also steadily increasing every year. A model projection by Kobayshi et al. (2015) suggested that total global seafood supplies will increase from 154 million tons in 2011 to 186 million tons by 2030 and aquaculture will be entirely responsible for the increase as conventional wild-capture fisheries have stagnated [4]. To sustain growing global aquaculture demand, it is important for the industry to have access to a wider range of new feed ingredient inputs, particularly dietary proteins and oils. Conventional fish meals and fish oils produced from a reduction in wild-caught forage fish have traditionally been the major sources of high-quality dietary proteins and oils in aquafeeds. However, these components have long since reached their plateau for economic and ecologically sustainable use by the industry [5,6]. Due to this fact, the aquaculture industry has already shifted to the use of plant-based proteins and oils (predominantly derived from soy, corn, wheat, and canola) to reduce the dependency on conventional fish meals and fish oils. The aquaculture industry is continuously seeking new reliable and inexpensive sources of proteins and oils to sustain the growing demand for aquaculture products [7].

In Canada, aquaculture production activities occur in every province and territory, and in 2017 alone contributed CAD 2.2 billion in GDP and employed 26,000 Canadians [8]. Plant-based proteins and oils, especially from soy, corn, wheat, and canola, are already in use as salmonid feed ingredients, but there is still a need for new ingredient sources to fill the gap [9]. Hemp processing by-products may have potential as new salmonid aquafeed ingredients due to their good profiles of essential amino acids and polyunsaturated fatty acid (PUFA) that are needed for fish growth and health [10,11]. While this may be the first effort with a focus on salmonids, encouraging results have been reported on the use of hemp proteins and oils for other commercially relevant aquaculture species such as sunshine bass, cobia, common carp, and Nile tilapia [12,13,14,15]. In our previous study, we reported on the characterization of triacylglycerols (TAGs) and other lipids derived from hemp by-products using UPLC-HRMS [16]. To further complete our research on hemp by-products valorization, we studied the protein content in hemp by-products, particularly hemp cake and hemp seed hulls in comparison with commercially available hemp hearts and hemp whole seeds. The objectives of this study were to report on the extraction and nutritional characterization of protein isolates (PIs) produced from hemp by-products and their antioxidant properties.

## 2. Results

### 2.1. Moisture, Ash, Carbohydrate, and Protein Content

The moisture and ash contents of hemp by-products together with whole hemp seeds and hemp hearts were measured. Hemp cake and hemp hulls had the highest moisture content at 8.2 and 6.7%, respectively. The whole hemp seeds had the lowest moisture content at only 3.1%. The ash content of the tested sample ranged from 2 to 6%. Hemp hulls had the lowest ash content, 2.4%, and hemp cake had the highest, 6.1%. The lipid content of all four hemp biomasses was reported previously [16]. The crude protein content was determined by nitrogen analysis using the average generalized nitrogen-to-protein conversion factor (N × 5.37) as determined by Gosukonda (2020) [17]. The protein content of hemp cake and hemp seed hulls was 30.4 and 8.6%, respectively. The highest protein content was observed in hemp hearts at 31.6% (Table 1). As expected, the values were increased in the defatted hemp cake and hemp hulls biomasses and were 31.6 and 16.6%, respectively (Figure 1). The carbohydrate contents of the hemp cake and hemp hulls were 21.3% and 33.7%, respectively, determined by a colorimetric assay using phenol and sulfuric acid as described by Dubois et al. (1956) [18].

### 2.2. Extraction of Protein Isolates (PIs) and Purity

Protein isolates (PIs) were extracted from defatted biomass of hemp cake and hemp hulls, together with whole hemp seeds and hemp hearts, using an alkaline extraction with different NaOH concentrations, i.e., 0.5 M, 0.1 M, and pH 10.0 followed by isoelectric precipitation using 1.0 M HCl. The percentage yields of PIs of all four defatted biomasses are shown in Figure 1 together with total crude protein values measured based on nitrogen analysis. The highest PI yield of the defatted biomass was observed for hemp hearts (32.3–42.9%), followed by whole hemp seeds (20.7–26.0%), apparent in all three alkaline conditions. The percentage yield of hemp cake ranged from 7.4 to 19.4%, whereas hemp seed hulls had the lowest PI yield range, from 7.9 to 9.3% of the defatted biomass (Figure 1). The PIs extracted using 0.1 M NaOH were selected for further biochemical characterization and in vitro digestibility tests. PI derived from hemp hearts was light yellow, whereas PIs extracted from hemp cake, hulls, and whole hemp seeds were light-to-dark brown in color (Figure 2).

### 2.3. Amino Acid (AA) Analysis

Waters AccQ-Tag Ultra method was used for amino acid (AA) analysis of PIs extracted from hemp seeds and the two by-products in three alkaline conditions, and the concentrations of 17 amino acids were determined [19]. The results are shown in Table 2 (PIs extracted at 0.1 M NaOH) and in Appendix A (PIs extracted at 0.5 M NaOH and pH 10.0), including all but one of the essential amino acids for fish; tryptophan (Trp) was not the part of the analysis. The total AA count of PIs extracted from hemp hulls ranged from 585.0 to 752.0 mg/g, which was relatively lower for all three alkaline conditions when compared to total AA count from PIs extracted from hemp cake and hemp seed biomasses. The total amino acid count in PIs extracted from hemp cake was in the range from 747.7 to 765.7 mg/g. Among individual amino acids, glutamic acid had the highest concentration, followed by arginine and aspartic acid in all PIs extracted from tested hemp biomasses, and their concentrations were in the ranges 120.3–174.0, 81.3–135.0, and 62.3–105.7 mg/g, respectively (Table 2 and Appendix A).

### 2.4. SDS-PAGE Gel Electrophoresis

The SDS-PAGE profiles of PIs extracted with 0.1 M NaOH derived from hemp hearts, whole seeds, and the two by-products are shown in Figure 3. The SDS-PAGE profiles of remaining PIs extracted at pH 10 and 0.5 M NaOH are shown in Appendix A. The profiles showed similar patterns within reducing and non-reducing conditions, respectively. Strong bands in the 30 kDa range point towards the presence of the acidic subunit (AS) of edestin, while dominant bands in the 20 kDa range are putatively the basic subunit of edestin and albumin (Figure 3) [20,21]. Bands in the 50 kDa range could be edestin, 7S vicilin-like protein, or β conglycinin [20,21]. Under reducing conditions, the bands of the subunits of edestin appear stronger than under non-reducing conditions. The identification of the bands is based on previously published data and is putative.

### 2.5. FT-IR Spectral Analysis of Biomass and Protein Isolates (PIs)

FT-IR spectra of all four PIs extracted at 0.1 M NaOH either from hemp by-products or from hemp seeds are shown in Appendix A. Amide I at 1630 cm^−1^ and amide II at 1520 cm^−1^ were the strongest peaks of all PIs. IR spectra of all four PIs showed almost identical peaks in FT-IR, suggesting similar composition. The FT-IR spectra of the hemp by-products and defatted biomasses are also shown in Appendix A.

### 2.6. Two-Phase In Vitro Gastric/Pancreatic Protein Digestibility (IVPD)

Protein digestibility of the PIs extracted using 0.1 M NaOH were further studied using two-phase in vitro gastric/pancreatic protein digestibility (IVPD). The highest digestibility was observed in the PI extracted from hemp hearts (98.5%), whereas the lowest was in the PI extracted from hemp hulls (87.8%). The digestibility of PIs derived from hemp cake and whole hemp seeds was 91.1 and 94.6%, respectively (Figure 4).

### 2.7. 2,2-Diphenyl-1-Picrylhydrazyl (DPPH) Radical Scavenging Activity

The PI extracted at 0.1 M NaOH from hemp by-products were tested for their DPPH radical scavenging activity according to the procedure described by Hatano et al. (1989) with minor modification [22]. All tested PIs showed weak but dose-dependent DPPH radical scavenging activity (Figure 5). At 1000 µg/mL concentration, PI derived from hemp cake and hemp hulls showed 40.4 and 58.6% of DPPH radical scavenging potency, respectively. The positive control used for the study (ascorbic acid) on the other hand possessed IC_50_ value of ~10 µg/mL.

### 2.8. Oxygen Radical Absorbance Capacities (ORAC) Assay

ORAC values of the PIs extracted from hemp by-products and hemp seeds were measured by the method described by Wu et al. (2004) [23]. The PI extracted from hemp hulls possessed the highest ORAC value of 1017.2 µM TE/g PI. Similarly, ORAC values of the PI extracted from hemp press-cake, hemp hearts, and hemp whole seed were 656.5, 719.7, and 677.4 µM TE/g PI, respectively (Figure 6).

### 2.9. Total Phenolic (TP) Content

The total phenolic content of the PIs extracted with 0.1 M NaOH was measured using the Folin–Ciocalteu method with 96-well plate format as described by Zhang et al. (2006) [24], and the results are shown in Figure 6. The PI extracted from hemp hulls possessed the highest phenolic content at 283 µM GAE/g. The total phenolic content of hemp cake PI was 192 µM GAE/g. The PI derived from hemp hearts had the lowest phenolic content at only 129.7 µM GAE/g, and whole hemp seeds had 185.0 µM GAE/g.

## 3. Discussion

Hemp (*Cannabis sativa* L.), an annual herbaceous plant that has been used as a source of food, fiber, and medicine for centuries [25], emerges as a potential source for value-added functional food ingredients and nutraceuticals [26]. Since its legalization in Canada, 77,800 acres of hemp were planted in 2018 alone, and the cultivation of industrial hemp is expected to rise significantly in the coming years based on increasing demand for hemp seed oil in the food sector and cannabidiol (CBD) containing hemp oil for medicinal purposes [27]. The production of large quantities of hemp oils will generate large volumes of by-products including hemp hulls, hemp cake (hemp press-cake or hemp meal), and other hemp plant parts. Finding high-value chemicals or utilizing hemp by-products in livestock feed including aquaculture industries will not only help to reduce the management burden of waste biomass but equally generates additional revenue for this emerging agricultural industry. With good nutritional value and digestibility, great interest has been drawn in hemp protein in both scientific and industrial fields [20].

In our previous study, we reported on the characterization of lipids extracted from hemp by-products for their possible feed application in the aquaculture industry [16]. Proteins from hemp by-products are another possible feed ingredient for livestock or aquaculture industries. The total protein content in the hemp by-products such as hemp cake was 31.6%, and only 8.6% for hemp hulls, determined by nitrogen analysis. The numbers were slightly lower than previously reported, which were 40.7 and 12.7% for hemp cake and hemp hulls, respectively [10]. This is may be due to differences in the cultivar or geographical conditions, and in the current study we used the average generalized N-to-P conversion factor N × 5.37 described by Gosukonda et al. (2020) [17] to determine crude protein content in hemp instead of the conventional conversion factor N × 6.25 used in the previous report [10,28]. The crude protein content increased to 31.6 and 16.6% in the defatted biomass (collected after hexane and 80% EtOH extraction) of hemp cake and hemp hulls, respectively (Figure 1). Among the tested hemp biomasses, not surprisingly, the defatted hemp hearts showed the highest protein content (58.8%).

Extraction efficiency of hemp by-products and hemp seeds under varying alkaline conditions were further studied. In a previous report, Potin et al. (2019) demonstrated that the extraction yield of hemp protein was significantly increased at pH 9 and higher [29]. Similarly, Dapčević-Hadnađev et al. (2018) studied the extraction efficiency of hemp proteins, comparing alkaline extraction/isoelectric precipitation with micellization procedures, and demonstrated that the alkaline extraction had better protein recovery [30]. In the current study, we examined the extraction efficiency at three different alkaline conditions using NaOH (0.5 M, 0.1 M, and pH 10.0), a common method used for plant seed protein extraction including hemp proteins [29,30,31,32]. At 0.1 M NaOH, slightly better extraction yields were observed for all tested defatted biomasses except hemp whole seeds (Figure 1). The extraction efficiency of PIs at 0.1 M NaOH for hemp cake, hemp hulls, hemp hearts, and hemp whole seeds was 61.4, 56.0, 67.7, and 69.6%, respectively, calculated based on the total crude protein content and the actual extraction yield. Even though extraction efficiency was slightly lower in hemp hearts as compared to whole hemp seeds at 0.5 M and 0.1 M NaOH concentrations, overall, the highest extraction yields were observed in hemp hearts in all three alkaline conditions, which was in line with a previous report by Shen et al. (2020). In that study, Shen et al. showed that the dehulling of hemp seeds significantly increased the extraction efficiency and the protein recovery yield without changing the composition of the PIs [28]. The PI extracted at 0.1 M NaOH from hemp press-cake was used for biochemical characterization for further pilot-scale extraction to study its nutritional value as a novel protein-rich ingredient for salmonid aquafeeds. This fish feeding study is currently in progress.

The storage protein edestin was reported as a major protein component in the hemp seed by Tang et al. (2006), where the physicochemical and functional properties of hemp protein isolates were compared with soy protein isolates [33]. In a later study, Memone et al. (2019) extracted PI from defatted hemp cake (hemp meal), resulting in nearly 86% protein content, constituted mainly by the storage protein edestin (70%) [30]. Edestin is a hexameric protein with each subunit consisting of two subunits that are described as the acidic subunit and the basic subunit [34]. The SDS-Page profiles in our study showed the presence of bands that could belong to edestin (~50 kDa) and two subunits of edestin (~30 kDa, ~20 kDa) [20,21]. The bands of the subunits appeared under both conditions but had stronger appearance under reducing conditions (Figure 3, Appendix A). Overall, the SDS-PAGE profiles revealed similar results for all analyzed hemp PIs except hemp hulls. Here, the lower solubility of the hemp hull PIs resulted in an overall lower protein content on the gel. Crude protein content in the PIs were determined by nitrogen analysis, which suggested that PIs extracted from all three alkaline conditions were enriched with >69.7% protein when the generalized N-to-P conversion factor of N × 5.37 was used for crude protein calculation (Table 2 and Appendix A). The lowest protein content was observed on PIs extracted from hemp hulls using 0.5 M NaOH, most probably due to pigmentation mostly by phenolic compounds as described by Potin et al. (2019) [29]. The protein components were almost identical for PIs derived either from hemp by-products or hemp seeds for both reducing and non-reducing conditions. This fact was further supported by the fact that FT-IR spectra of all PIs isolated from hemp by-products or hemp seeds with 0.1 M NaOH were identical, suggesting that the protein composition should be similar. All PIs showed two strong absorption peaks around 1630 cm^−1^ and 1520 cm^−1^ belonging to amide I and II. The FT-IR spectra of original biomasses (Appendix A), on the other hand, showed another strong absorption peak at 1740 cm^−1^ belonging to an ester bond of lipid, especially triacylglycerols (TAGs), containing a significant amount in the lipid [16]. Moreover, similar HPLC profile was observed for all PIs extracted from hemp seeds and hemp by-products (Appendix A).

Hemp protein isolate showed a high degree of digestibility when studied using a static model of gastrointestinal digestion (GID) [32]. In the current study, we have performed in vitro digestibility on all four PIs derived from hemp by-products and hemp seeds extracted at 0.1 M NaOH. The percentage digestibility was estimated using two-phase in vitro gastric/pancreatic protein digestibility (IVPD) following Yegani et al. (2013) [35], with modifications to optimize for salmonids according to Tibbetts at al. (2020) [36]. The PI derived from hemp hearts having 85.1% purity based on nitrogen analysis showed highest IVPD (98.5%). Even though PIs derived from hemp cake had 87.8% purity, the IVPD was lower, but still very high (91.1%), which is highly similar to the IVPD of 88–91% reported previously for hemp cake PIs by Wang et al. (2008) [32]. The PI extracted from hemp hulls showed the lowest purity (77.0%) and also had the lowest IVPD (87.8%). These results clearly suggest that the PIs derived from hemp by-products and hemp whole seeds containing hulls diminish their digestibility somewhat, which may be due to presence of pigmentation. The darker color of the PIs extracted from hemp by-products compared to hemp hearts also indicated the presence of phenolic compounds in PIs derived from by-products (Figure 2). Quality of protein is not only defined by its purity, but it is also important to have an appropriate essential AA profile and high digestibility. The digestibility of PIs derived from either hemp cake or hemp hulls was above 87.8%, which is excellent when compared with other conventionally used protein-rich salmonid aquafeed ingredients. For instance, the range of IVPD determined in this study for hemp PIs (88–98%) is consistent with or higher than those measured in vivo for premium fish meals (83–95%), poultry by-product meal (74–94%), corn gluten meal (92%), corn protein concentrate (91%), soybean meal (77–94%), and soy protein concentrate (90%) [37,38,39,40].

Macronutrient composition and protein quality of hemp seeds and products derived from hemp seeds grown in western Canada were well studied by House et al. (2010) [10]. The hemp seeds and other biomasses were directly analyzed for their AA content and protein digestibility-corrected amino acid scores (PDCAAS), measured using an in vivo rat bioassay. The results suggested that hemp protein contains all ten essential AAs required for fish and protein digestibility ranges from 84.1 to 97.5%. In our study, we also observed similar AA profiles for all tested hemp PIs except tryptophan (Trp), which was not determined. It is worth mentioning here that this is the first report of the extraction of PI from hemp hulls and its biochemical characterization, while Wang et al. (2008) first reported on this for PIs produced from hemp cake [32]. Even though all tested essential AAs were detected in PIs extracted from hemp by-products, a significant difference in total AA count was observed between PIs extracted from hemp by-products and hemp seeds. The highest total AA content observed for hemp heart PI ranged from 796.7 to 904.0 mg/g and PIs extracted from hemp hulls ranged from 585.0 to 752.0 mg/g. The total AA content of hemp cake PIs extracted at three different concentrations of NaOH ranged from 747.7 to 765.7 mg/g, suggesting that the presence of hulls reduces the total AAs. Considering individual AAs, glutamate (Glu), arginine (Arg), and aspartate (Asp) are the dominant AAs found in hemp PIs either extracted from hemp by-products or hemp seeds.

Free radicals are generated during normal cellular metabolism, and overproduction of such highly reactive radicals, including reactive oxygen species (ROS) and reactive nitrogen species (RNS), has a negative impact on the health of both humans and animals [41]. ROS are associated with various chronic diseases such as cancer, neurodegenerative disease, and respiratory diseases [42]. Antioxidants such as tocopherol (vitamin E), ascorbic acid (vitamin C), β-carotene, and other plant-derived metabolites help to reduce such negative impact or defend against free radical damage [43]. There is a report on hempseed protein hydrolysates (HPHs) exhibiting antioxidant activities and antiproliferative effects on cancer HeLa cell growth [44]. In this regard, we further tested the PIs extracted from hemp by-products and hemp seeds at 0.1 M NaOH for their DPPH radical scavenging activity and measured their ORAC values. The tested PIs showed weak DPPH radical scavenging properties but possessed dose-dependent radical scavenging activity. PI extracted from whole hemp seeds showed the highest DPPH radical scavenging activity of 71.3% at 1000 µg/mL concentration, only in the roasted hemp sample. Among the remaining three hemp samples received from the same industrial collaborator, hemp hull PI had the highest DPPH radical scavenging properties. Similarly, the ORAC value of the PI extracted from hemp hulls had the highest among the tested PIs (i.e., 1017.2 µM TE/g). The ORAC also correlated with the total phenolic content, i.e., PIs extracted from hemp hulls had the highest total phenolic content at 283.0 mm GAE/g. These results clearly suggest that the ORAC values of these PIs may be due to pigmentation by phenolic compounds.

In conclusion, we have extracted PIs from hemp by-products and studied their biochemical characterization, in vitro protein digestibility, and antioxidant potency. To the best of our knowledge, this is the first report on the extraction and biochemical characterization of PI from hemp hulls. Even though the total protein content of hemp press-cake and hemp hulls defatted biomass was 31.6 and 16.6%, actual PIs yields were 19.4 and 9.3%, respectively. The extraction efficiency of protein at 0.1 M NaOH was 61.4% for hemp press-cake and 56.0% for hemp hulls. FT-IR, HPLC, and the SDS-PAGE gel electrophoresis results suggested that PIs extracted either from hemp by-products or from hemp seeds have a similar composition. Storage protein edestin was the dominant protein in the PIs extracted from hemp seeds and hemp by-products at 0.1 M NaOH. Glutamate (Glu), aspartate (Asp), and arginine (Arg) are the three major AAs present in PIs extracted from hemp seed and by-products. The PIs extracted from hemp hulls showed the strongest antioxidant properties and correlate with phenolic content.

## 4. Materials and Methods

### 4.1. Research Materials

Research materials, hemp hulls, hemp hearts, and hemp press-cake, were provided by Hemp Oil Canada (Ste. Agathe, MB, Canada). Roasted whole hemp seeds were purchased from a local market (Bulk Barn, Halifax, NS, Canada). All samples were stored at room temperature prior to extraction and PIs were stored frozen (−20 °C). IR spectrum was recorded in a Nicolet 1S10 FT-IR spectrometer (Thermo Fisher Scientific, Waltham, MA, USA). HPLC was carried out on an Agilent 1100 Series HPLC equipped with a diode array detector (Agilent Technologies, Santa Clara, CA, USA). All chemical reagents were purchased from Sigma-Aldrich (St. Louis, MO, USA) unless written otherwise.

### 4.2. Moisture, Ash, and Carbohydrate Content

The moisture content of pulverized hemp hearts, whole seed, hemp cake, and hemp seed hulls was determined using an HE53 Moisture Analyzer (Mettler Toledo, Greifensee, Switzerland). The pulverized samples were further used for ash content determination by heating to 550 °C for 18 h in a Muffle furnace. The remaining ash was slowly cooled to RT in a desiccator and its mass was used to calculate the ash content percentage. The carbohydrate content of the pulverized samples was determined by colorimetric assay as follows. A 10 mg measure of biomass was hydrolyzed with 2.5 M HCl for 3 h at 90 °C in a water bath. Acid was neutralized using sodium carbonate and samples were centrifuged. An aliquot of supernatant (125 µL) was mixed with 875 µL of Milli-Q water and 1 mL of 5% Phenol solution was added. Concentrated sulfuric acid (5 mL) was added and samples were incubated for 20 min at 30 °C. Absorbance of samples was measured at 490 nm using a GENESYS 10S UV–Vis Spectrophotometer (Thermo Scientific, MA, USA). Samples of dextrose (0.0625–2.0 mg/mL) were subjected to the same procedure and used as an external standard for quantification.

### 4.3. Crude Protein Content Determined by Nitrogen Analysis

Total nitrogen (N) contents were determined by elemental analysis (950 °C furnace) using a N analyzer (model FP-828P, LECO Corporation, St. Joseph, MI) calibrated with EDTA (LCRM^®^, Cat. #502-896), ultra-high purity oxygen as the combustion gas, and ultra-high purity helium as the carrier gas. Crude protein (CP) contents (as-is basis) were estimated using the average generalized hemp N-to-P conversion factor (N × 5.37) as described in the literature [17]. The extraction efficiency of protein was calculated using the formula [Extraction efficiency (%) = (Percentage yield of PI/Percentage crude protein content) × 100].

### 4.4. Extraction of Protein Isolates (PIs)

The pulverized biomass of the by-products, hemp seed hearts, and whole hemp seeds was defatted by extracting lipid with hexane at room temperature followed by EtOH at 60 °C. The resulting defatted biomasses were subjected to protein extraction according to the well-known alkaline extraction process used for the extraction of soybean protein isolate, with minor modification at three different NaOH concentrations [32]. In brief, defatted biomass (1–100 g) was mixed with 0.5 M NaOH, 0.1 M NaOH, and at pH 10.0 with 1 M NaOH (*w*/*v*, 1:10) and stirred overnight at room temperature. The extraction mixture was centrifuged at 12,000 rpm for 10 min, the supernatant was collected, and the pH was adjusted to 5.0 using 1 M HCl. The precipitate was collected by centrifugation at 4000 rpm for 20 min and suspended in a small volume of Milli-Q water and centrifuged at 4000 rpm for 20 min. The final participate was collected, suspended into Milli-Q water, had pH adjusted to 7.0 by 1 M NaOH, and was freeze-dried. The lyophilized PI sample was stored frozen (−20 °C).

### 4.5. Amino Acid Analysis

Amino acid analysis of the protein isolated was conducted according to Waters AccQ-Tag Ultra (Waters Milford MA, USA) method [19]. In brief, 10 mg of protein isolate containing internal standard α-aminobutyric acid (ABA) was hydrolyzed with 1 mL of constant boiling HCl (~12 M) at 120 °C for 24 h in triplicate. The reaction was cooled down to room temperature, filtered through cotton, and diluted to 5 mL with Milli-Q water. Samples were further filtered through a 0.22 µm membrane filter. The hydrolyzed solution (20µL) was dried under vacuum. The dried hydrolyzed sample was derivatized according to the Waters AccQ-Tag Ultra protocol as follows: sample was dissolved in 80 µL of borate buffer (0.2 M, pH 8.8) and 20 µL of Waters AccQ-Tag Ultra derivatization reagent was added. The mixture was vortexed and heated at 55 °C for 10 min and subjected to UPLC analysis. Waters amino acid hydrolysate standard was derivatized in the same way and used as an external standard for quantification. UPLC analysis was performed on an Agilent 1260 LC system equipped with a UV DAD monitoring system at 260 nm. The amino acid separation was carried out using a Waters AccQ-Tag Ultra C18 column (1.7 µm, 2.1 × 100 mm). The solvent system consisted of: A: AccQ-Tag Ultra eluent A (5% *v*/*v* in water) and B: AccQ-Tag Ultra Eluent B. The flow rate was 0.7 mL/min and the column temperature was maintained at 55 °C. Injection volume was 1 µL. The following solvent gradient was used: 0–1.54 min 99.9% A—0.1% B; 6.74 min, 90.9% A—9.1% B; 8.74 min, 78.8% A—21.2% B; 9.04–9.64 min, 40.4% A—59.6% B; 9.73–12.00 min, 99.9% A—0.1% B. Agilent ChemStation software was used for data acquisition and analysis.

### 4.6. SDS-PAGE Gel Electrophoresis

PIs extracted from hemp by-products and hemp seeds were dissolved in lysis buffer at 4 mg/mL containing 50 mM Tris-HCl at pH 7.4, 150 mM NaCl, 1% Triton x-100 (*v*/*v*), and 0.1% SDS (*w*/*v*). PIs showed varying degrees of solubility and the resulting PI solutions were centrifuged at 13,000 rpm for 5 min and the supernatant (30 µL) was mixed with 4× Laemmli buffer (10 µL, Bio-Rad, Hercules, CA, USA) with and without reducing agent dithiothreitol 2% (*w*/*v*), and incubated at 90 °C for 5 min in a water bath. Samples (20 µL) were then deposited in the wells of an SDS-PAGE gel (12% acrylamide) and run in 10× Tris/Glycine/SDS buffer (Bio-Rad, Hercules, CA, USA) at 150 v until the tracking dye reached the bottom of the gel. The gel was dyed and de-stained using Bio-Safe Coomassie G-250 stain (Bio-Rad, Hercules, CA, USA) according to manufacturer’s instruction. The final image of the gel was recorded using a Molecular Imager ChemiDOC XRS with Image Lab Software (Bio-Rad, Hercules, CA, USA).

### 4.7. Two-Phase In Vitro Gastric/Pancreatic Protein Digestibility (IVPD)

Protein digestibility of the four PIs was estimated using a two-phase in vitro gastric/pancreatic protein digestibility (IVPD) assay following Yegani et al. (2013), with modifications to optimize for salmonids according to Tibbetts et al. (2020) [35,36]. In brief, IVPD was measured by incubation of 250 mg of the test sample in porcine pepsin (P7000, Sigma-Aldrich) enzyme solution (25 mg/mL *w*/*v* in 0.2 N HCl at pH 1.0) for 4.5 h at 39 °C (*gastric phase*) with head-over-heels agitation on a Tube Rotator (model 13916-822, VWR Canada) equipped with Rotisserie Assembly for 50 mL tubes (model 13916-834, VWR Canada). Samples were then incubated in the same manner in porcine pancreatin containing amylase, lipase, and protease (P1750, Sigma-Aldrich) enzyme solution (100 mg mL^−1^
*w*/*v* in 0.05 M Tris, 0.0115 M CaCl_2_ buffer at pH 8.0) for 9 h (*pancreatic phase*) under the same incubation conditions. Assays were conducted with four analytical replicates (n = 4) per test PI sample and each was run in parallel with procedural blanks containing no sample. The CP contents of the hemp PI samples were normalized to 100% dry matter basis by drying triplicate aliquots of each in an oven at 105 °C for 18 h. IVPD of each sample was then calculated as:$$\mathrm{IVPD}\;\left(\%\right)=\frac{\left(\mathrm{CP}\;\mathrm{in}\;\mathrm{test}\;\mathrm{PI}\;\mathrm{sample}\right)}{\left(\mathrm{CP}\;\mathrm{in}\;\mathrm{digested}\;\mathrm{residue}-\mathrm{CP}\;\mathrm{in}\;\mathrm{blank}\right)}\;÷\mathrm{CP}\;\mathrm{in}\;\mathrm{test}\;\mathrm{PI}\;\mathrm{sample}\;\times 100\%$$

IVPD results are reported as mean ± standard deviation and were statistically compared by one-way analysis of variance (ANOVA) using SigmaStat^®^ software (v.3.5) and a 5% level of probability (*p* < 0.05) to sufficiently demonstrate a statistically significant difference. Where significant differences were observed, treatment means were differentiated using pairwise comparisons using the Tukey test. Raw data were checked for goodness-of-fit using the Kolmogorov–Smirnov tests (SigmaStat^®^ v.3.5) for normality (*p* = 0.647, passed) and equal variance (*p* = 0.319, passed).

### 4.8. 2,2-Diphenyl-1-Picrylhydrazyl (DPPH) Radical Scavenging Activity

DPPH radical scavenging activity was analyzed according to the procedure described by Hatano et al. (1989), with minor modification [22]. In brief, 100 µL of extract at various concentrations in phosphate buffered saline (pH 7) was mixed with equal volume of 60 µM DPPH solution in 75% EtOH, and the resulting solution was thoroughly mixed and absorbance measured at 520 nm after 30 min using a BioTek Cytation 5 imaging reader (BioTek, Winooski, VT, USA). The scavenging activity was determined by comparing the absorbance with that of controls containing only DPPH and 75% EtOH. Vitamin C, a known antioxidant, was used as a positive control having IC_50_ value of ~10 ug/mL. Measurements were carried out in triplicate.

### 4.9. Oxygen Radical Absorbance Capacity (ORAC) Assay

The ORAC assay was performed as described previously by Wu et al. (2004), with modification [23]. The assay is based on the principle that a fluorescent probe is oxidized by the addition of a free radical generator (APPH) which quenches the fluorescent probe over time. Antioxidants present in the sample block the generation of free radicals until the antioxidant activity of the sample is depleted. PIs were dissolved in 0.1 M NaOH solution 2 mg/mL and further diluted 80 times by Milli-Q water to obtain stock solution. For ORAC, 25 μL of sample (stock solution) or standard (Trolox) diluted in 75 mM phosphate buffer (pH 7.4) was added to wells, followed by the addition of 150 μL of fluorescein (200 nM). Plates were incubated at 37 °C for 10 min and the reaction was initiated by addition of 25 μL of 150 mM AAPH. Fluorescence decay was monitored using a fluorescence plate reader (SpectraMax Gemini XS, Molecular Devices, San Jose, CA, USA) at an excitation wavelength of 485 nm and emission wavelength of 520 nm at 37 °C. Each extract was tested in triplicate at three concentrations. Trolox, a water-soluble vitamin E analog, was used as the calibration standard and the results were expressed as Trolox equivalents (TE).

### 4.10. Total Phenolic (TP) Content

Total phenolic (TP) content of the protein isolate was determined based on a Folin–Ciocalteu (FC) method adapted to a 96-well microplate format (Zhang et al., 2006), with minor modifications [24]. In brief, PIs were dissolved in 0.1 M NaOH solution with 2.0 mg/mL concentration, diluted 80 times by Milli-Q water to obtain stock solution, and 20 µL of the sample (stock solution) or standard was mixed with 40 µL of FC reagent (10%). After 5 min, 160 µL of 700 mM sodium carbonate was added and incubated for 2 h at room temperature. Absorbance was measured at 765 nm using a plate reader SpectraMax Plus (Molecular Devices). Gallic acid was used to generate a calibration curve, and the total phenolic content was expressed as gallic acid equivalents (GAE)/g protein isolate.

## Figures and Tables

**Figure 1 molecules-27-04794-f001:**
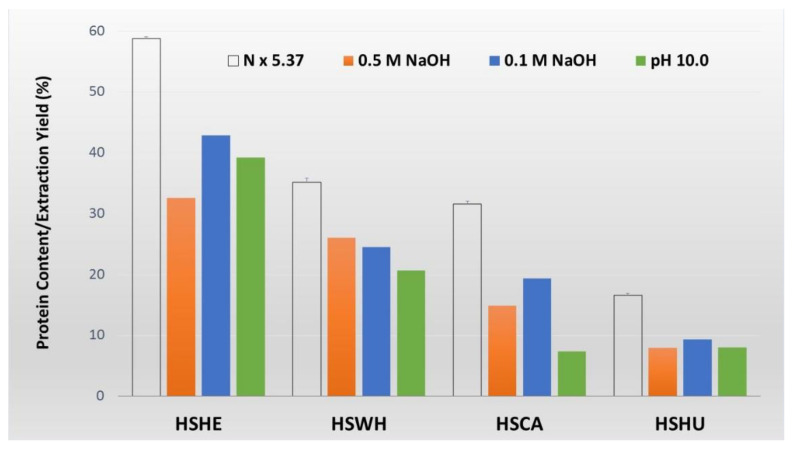
Crude protein content and extraction yield of protein isolates from defatted hemp seeds and defatted by-products. HSHE—hemp hearts; HSWH—hemp whole seed; HSCA—hemp cake; HSHU—hemp seed hulls.

**Figure 2 molecules-27-04794-f002:**
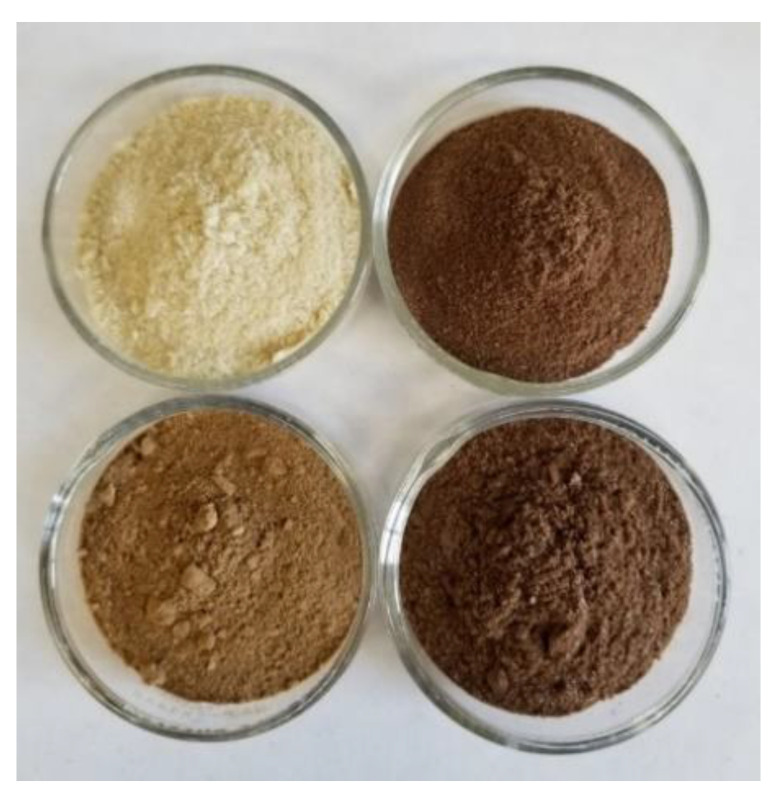
Protein isolate extracted from hemp seeds and hemp by-products. Clockwise from top left: protein isolate from hemp hearts, hemp whole seed, hemp hulls, and hemp cake.

**Figure 3 molecules-27-04794-f003:**
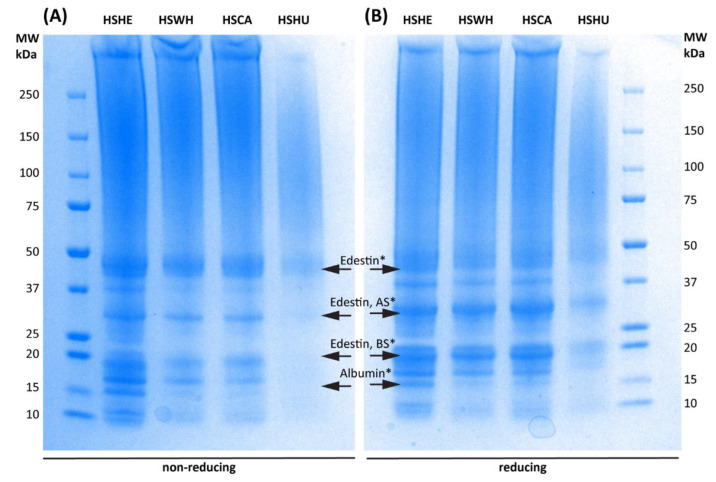
Non-reducing (**A**) and reducing (**B**) SDS-PAGE profiles for protein isolates extracted from hemp seeds and hemp by-products at 0.1 M NaOH. HSHE—hemp hearts; HSWH—hemp whole seed; HSCA—hemp cake; HSHU—hemp seed hulls. * Based on previously published data from Wang et al. (2019) [20] and Pavlovic et al. (2019) [21].

**Figure 4 molecules-27-04794-f004:**
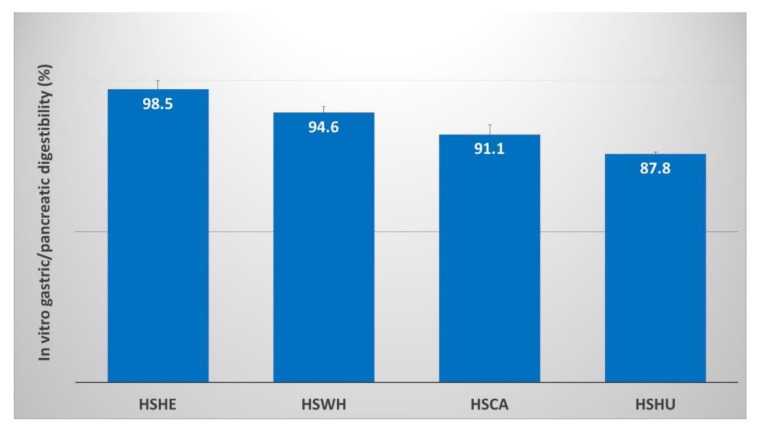
Two-phase in vitro gastric/pancreatic protein digestibility (IVPD) of protein isolates (PIs) extracted from hemp seed and by-products. Data are expressed as the means ± SD (n = 4). HSHE—hemp hearts; HSWH—hemp whole seed; HSCA—hemp cake; HSHU—hemp seed hulls.

**Figure 5 molecules-27-04794-f005:**
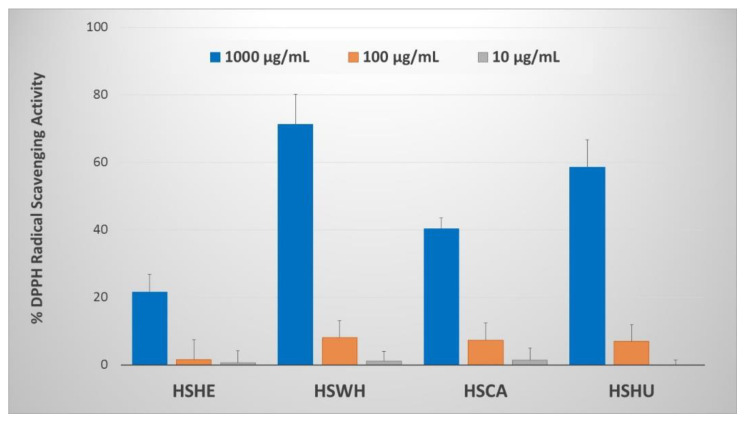
DPPH radical scavenging activity of protein isolates extracted from hemp seed and by-products. Data are expressed as the means ± SD (n = 3). HSHE—hemp hearts; HSWH—hemp whole seed; HSCA—hemp cake; HSHU—hemp seed hulls.

**Figure 6 molecules-27-04794-f006:**
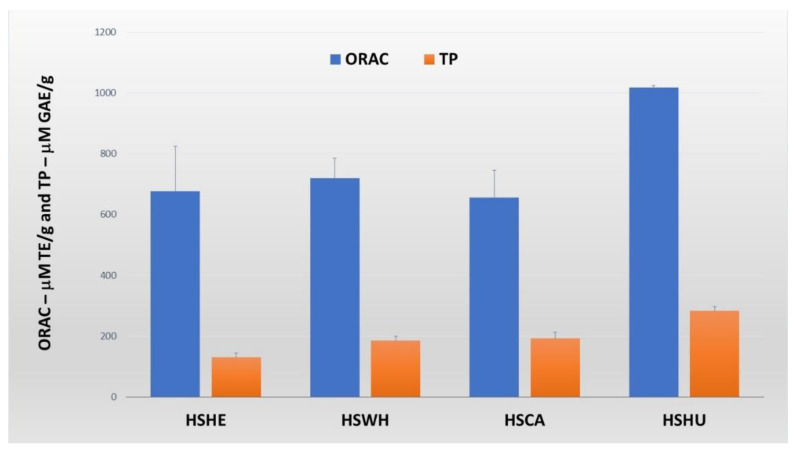
ORAC values and total phenolic (TP) content of protein isolates extracted from hemp seed and by-products. Data are expressed as the means ± SD (n = 4 for ORAC and n = 3 for TP). HSHE—hemp hearts; HSWH—hemp whole seed; HSCA—hemp cake; HSHU—hemp seed hulls.

**Table 1 molecules-27-04794-t001:** Moisture, ash, carbohydrate, and protein content of hemp hearts, whole seeds, hemp cake, and hemp seed hulls.

Content/Sample	HSHE	HSWH	HSCA	HSHU
Moisture (%)	5.1 ± 0.1	3.1 ± 0.0	8.2 ± 0.1	6.7 ± 0.0
Ash (%)	5.3 ± 0.5	4.5 ± 0.2	6.1 ± 0.2	2.4 ± 0.0
Carbohydrate (%)	2.8 ± 0.2	11.1 ± 0.5	21.3 ± 6.3	33.7 ± 6.9
Lipid (%) ^1^	54.7 ± 2.3	48.0 ± 2.8	13.1 ± 0.3	17.5 ± 0.1
Crude Protein (% N × 5.37)	31.6 ± 0.2	27.1 ± 0.2	30.4 ± 0.5	8.6 ± 0.1

^1^ Data were from literature Banskota et al. (2022) [16]. Data are expressed as the means ± SD (n = 3). HSHE—hemp hearts; HSWH—hemp whole seed; HSCA—hemp cake; HSHU—hemp seed hulls.

**Table 2 molecules-27-04794-t002:** Amino acid analysis of hemp protein isolates extracted at 0.1 M NaOH (results expressed in mg/g).

Amino Acid/Sample	HSHE	HSWH **	HSCA	HSHU
Histidine (His) *	23.7 ± 1.5	25.0 ± 2.8	23.7 ± 2.1	18.0 ± 0.0
Serine (Ser)	12.3 ± 3.5	10.0 ± 0.0	13.7 ± 3.8	8.0 ± 0.0
Arginine (Arg) *	112.0 ± 6.6	115.0 ± 8.5	109.0 ± 7.9	81.3 ± 1.2
Glycine (Gly)	37.7 ± 2.5	39.5 ± 3.5	38.0 ± 2.6	29.7 ± 0.6
Aspartate (Asp)	98.0 ± 5.6	95.5 ± 4.9	88.7 ± 1.2	72.3 ± 0.6
Glutamate (Glu)	156.0 ± 9.2	159.0 ± 4.2	146.7 ± 5.5	117.0 ± 0.0
Threonine (Thr) *	16.3 ± 2.1	15.0 ± 0.0	17.7 ± 2.9	12.0 ± 0.0
Alanine (Ala)	37.7 ± 2.5	38.0 ± 1.4	34.7 ± 1.2	28.0 ± 0.0
Proline (Pro)	32.3 ± 2.1	33.5 ± 0.7	32.0 ± 1.7	25.0 ± 0.0
Cysteine (Cys)	-	-	1.0 ± 1.0	-
Lysine (Lys) *	26.3 ± 2.1	24.0 ± 2.8	22.7 ± 1.2	19.3 ± 0.6
Tyrosine (Tyr)	22.0 ± 3.5	24.0 ± 2.8	18.0 ± 3.5	17.0 ± 1.0
Methionine (Met) *	24.0 ± 3.6	27.0 ± 0.0	19.0 ± 6.1	17.7 ± 2.9
Valine (Val) *	51.3 ± 3.1	53.5 ± 0.7	48.7 ± 2.1	39.0 ± 0.0
Isoleucine (Ile) *	42.3 ± 3.1	43.5 ± 0.7	40.0 ± 1.7	32.0 ± 0.0
Leucine (Leu) *	61.3 ± 4.0	63.5 ± 0.7	58.0 ± 2.6	46.0 ± 0.0
Phenylalanine (Phe) *	43.3 ± 3.1	44.5 ± 6.4	42.0 ± 4.4	32.3 ± 0.6
Total Amino Acid (mg/g)	796.7 ± 48.0	810.5 ± 13.4	753.3 ± 30.6	594.7 ± 1.5
Crude Protein (% N × 5.37)	85.1 ± 0.2	86.4 ± 0.1	87.8 ± 0.3	77.0 ± 0.4

* Essential amino acids for fish, ** data from duplicate experiment, (-) not detected, HSHE—hemp hearts; HSWH—hemp whole seed; HSCA—hemp cake; HSHU—hemp seed hulls.

## Data Availability

Not applicable.

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
