# Peer review of "Biochemical Characterization and In Vitro Digestibility of Protein Isolates from Hemp (Cannabis sativa L.) By-Products for Salmonid Feed Applications"

_molecules, 2022, doi:10.3390/molecules27154794_

Round 1
Reviewer 1 Report
At 128 row you have to write the title of the table; now is written This is a table. Tables should be placed in the main text near to the first time they are cited.I recommend to replace this with the title of the table
Author Response
Thank you. The title of Table 2 was added to the revised manuscript.
Reviewer 2 Report
the paper is interesting and the topic fit with aim and scope of the Journal. In my opinion the paper can be accepted pending revisions, particularly:
- add in figures 1, 6 the standard deviations
- i suggest to the Authors to add also HPLC analyses of the extracts in order to obtain deep info on the reproducibility of the procedure and the quality control of the extracts
Author Response
We appreciate the positive comments. The following changes were made in the revised manuscript based on the recommendations.
- Figure 1 and 6 were revised having standard deviations for crude protein content, ORAC and total phenolic (TP) values. Unfortunately, most of the protein extraction was performed on a single experiment on a scale of 50-100 g defatted biomass, and the percentage yield of PIs was presented without standard deviation.
- HPLC analysis of protein isolates (PIs) was done using the ACE C4 column and the HPLC chromatograms were added in the supplementary document (Figure S4). No significant difference was observed on the HPLC chromatogram of PIs extracted from hemp seeds and hemp seed by-products.
Reviewer 3 Report
The objective of this work (manuscript molecules-1827531) is the revalorization of some of the by-products of hemp as a protein source for aquaculture industry. For this, the authors have carried out the study of their protein content as well as the amino acid composition, in vitro digestibility, and antioxidant activity (DPPH and ORAC) of protein isolates extracted (PIs) from them. The work does not present great novelties, since the hemp species, including many of its by-products, has been extensively studied. The main contribution of the work is the characterization of the protein extracts obtained from the hemp seed hulls.
In my opinion the work deserves its publication in Molecules only after major revision.
. Comments to the Authors
SDS-PAGE gel electrophoresis. Hemp seed protein consists mainly of globulin (edestin) and albumin (Wang, Q.; Xiong, Y.L. Processing, Nutrition, and Functionality of Hempseed Protein: a review. Compr. Rev. Food Sci. Food Saf. 552 2019, 18, 936-952.; Potin, F., Saurel, R. (2020). Hemp seeds as a source of dietary protein. In: Crini, G., Lichtfouse, E. (eds) Sustainable Agriculture Reviews 42. Reviews of sustainable agriculture, vol 42 Springer, Cham. https://doi.org/10.1007/978-3-030-41384-2_9). However, the authors have not identified albumin and have strangely identified tubulin. To my knowledge, tubulin has not previously been identified in hemp products. However, the authors state that the identification of the bands has been made based on previously published data. Surprisingly, the authors also make no reference to the identification of tubulin in either Section 2.4 or Section 3.
Sections 2.2. and 2.3. What is the difference between the ‘crude protein content’ reported in these Sections. I understand that in Section 2.2 the ‘crude protein content’ (Figure 1) is determined indirectly by using a N-to-protein conversion factor that is computed based on sums of amino acids and in Section 2.3. it is assessed directly by analyzing amino acids quantitatively. But, both determinations are named in the same way and there is no explanation in the text of how 'crude protein' was calculated in Table 2. In addition, Table 2 header is missing.
Section 4.3 and Table 2. Why do the authors use the generalized hemp N-to-P conversion factor 5.37? They state that this value is described in literature [17,44]. However, in reference 17 a conversion factor 6.25 is used. On the other hand, does the conversion factor 5.37 that the authors use come from calculating the mean value of the maximum and minimum values proposed for the conversion factor in reference 44? If so, it is not obvious at first glance and the authors should clarify this in the text.
. Other comments
Abstract. The authors should use here the generalized N-to-P conversion factor of hemp instead of the conventional one (6.25), since it does not overestimate its real protein content.
Section 2.1, lines 75-76. The phrase is wrong. Please replace it with: ‘The protein content of hemp cake and hemp seed hulls were 35.4 and 10.0%, respectively’.
Section 2.2. The reported value for extraction efficiency in whole hemp seeds when using pH 10.0 (20.7%) does not seem to match that shown in the bar chart. Please review it.
Author Response
Thank you for your thorough review of the manuscript and valuable comments. The following changes were made to the revised manuscript.
- Regarding to SDS-PAGE gel protein identification, it was our typing mistake in Figure 3 that we wrote tubulin instead of albumin. In section 2.4 we clearly described the presence of albumin. Figure 3 was revised accordingly. We appreciate identifying the miss identification of the protein in Figure 3.
- Crude protein content was calculated based on N-to-P conversion factor. The amino acid analysis, on the other hand, will give true protein content. Unfortunately, we didn’t analyzed all amino acids, thus we simply use the term “total amino acid count” in section 2.3. A statement describing the protein content of PIs based on nitrogen analysis was removed from section 2.3 to avoid confusion. Title of Table 2 was added.
- To avoid overestimation of protein content in hemp seeds, hemp seed by-products, defatted biomasses or in PIs, crude protein content was calculated using the average generalized N-to-P conversion factor (Nx5.37) as determined by Gosukonda et al. 2020 (ref 17 in revised manuscript). Extraction efficiency and reference numbers were changed accordingly. The crude protein content described in Table 2 calculated using N-to-P conversion factor (Nx6.25) were removed. A new statement was added in the results and discussion section describing the N-to-P conversion factors. An additional statement was added in the discussion section describing the extraction efficiency and extraction yields.
- Generalized N-to-P conversion factor of hemp (Nx5.37) was used to calculate protein Revised both abstract and main text accordingly.
- Section 2.1, lines 75-76. The phrase was corrected as “The protein content of hemp cake and hemp seed hulls were 30.4 and 8.6%, respectively”. New numbers were calculated based on the generalized N-to-P conversion factor of hemp (Nx5.37).
- Figure 6 was revised by adding standard deviations plus 2D-diagram reflecting the correct value for extraction yield in whole hemp seeds when using pH 10.0 (20.7%).
Round 2
Reviewer 2 Report
the revised paper reports all changes required and now can be accepted for publication
Reviewer 3 Report
Thank you very much for heeding the recommendations in my report. In my opinion, the work deserves its publication in the Journal Molecules in its current format.